# Visualizing the 4D Impact of Gold Nanoparticles on DNA

**DOI:** 10.3390/ijms25010542

**Published:** 2023-12-30

**Authors:** Hosam Abdelhady, Fadilah Aleanizy, Fulwah Alqahtani, Abdullah Bukhari, Sahar Soliman, Samaresh Sau, Arun Iyer

**Affiliations:** 1Department of Physiology and Pharmacology, College of Osteopathic Medicine, Sam Houston State University, Conroe, TX 77304, USA; 2Department of Pharmaceutics, College of Pharmacy, King Saud University, Riyadh 11451, Saudi Arabia; 3College of Medicine, Taibah University, Medina 41477, Saudi Arabia; 4Department of Pharmaceutical Sciences, Eugene Applebaum College of Pharmacy & Health Sciences, Wayne State University, Detroit, MI 48201, USA

**Keywords:** atomic force microscope, AFM, ROS, genotoxicity, gold nanoparticles, AuNPs, cancer, COVID-19, Lewis acid, biosensors

## Abstract

The genotoxicity of AuNPs has sparked a scientific debate, with one perspective attributing it to direct DNA damage and another to oxidative damage through reactive oxygen species (ROS) activation. This controversy poses challenges for the widespread use of AuNPs in biomedical applications. To address this debate, we employed four-dimensional atomic force microscopy (4DAFM) to examine the ability of AuNPs to damage DNA in vitro in the absence of ROS. To further examine whether the size and chemical coupling of these AuNPs are properties that control their toxicity, we exposed individual DNA molecules to three different types of AuNPs: small (average diameter = 10 nm), large (average diameter = 22 nm), and large conjugated (average diameter = 39 nm) AuNPs. We found that all types of AuNPs caused rapid (within minutes) and direct damage to the DNA molecules without the involvement of ROS. This research holds significant promise for advancing nanomedicines in diverse areas like viral therapy (including COVID-19), cancer treatment, and biosensor development for detecting DNA damage or mutations by resolving the ongoing debate regarding the genotoxicity mechanism. Moreover, it actively contributes to the continuous endeavors aimed at fully harnessing the capabilities of AuNPs across diverse biomedical fields, promising transformative healthcare solutions.

## 1. Introduction

Gold nanoparticles (AuNPs) have been found to induce RNA and DNA damage, leading to viral and cellular genotoxicities [1,2,3,4]. These properties, along with their ultra-small sizes, high surface areas, and adjustable surface chemistry, make them ideal candidates for many biomedical applications such as biosensors, drug targeting, molecular imaging, photothermal therapy, and the treatment of viral infections [5,6,7]. However, the actual nucleic acid damage mechanism(s) of AuNPs is not fully understood. While some studies suggest that AuNPs generate free radicals through Fenton-type reactions, leading to significant DNA damage and apoptosis [3,8,9,10,11], other studies have demonstrated the direct toxicity of AuNPs independent of reactive oxygen species (ROS) [12], leading to structural changes, physical damage, or other mechanisms that compromise the integrity of the DNA molecule. These conflicting results regarding the genotoxic mechanisms of AuNPs emphasize the importance of studying the real-time interactions between AuNPs and individual DNA molecules in a near-molecular environment. Imaging techniques that provide high-resolution and real-time visualization, such as atomic force microscopy (AFM) and other advanced microscopy methods, can be invaluable for studying the dynamic processes occurring at the molecular level. To this end, we applied the four-dimension atomic force microscope (4DAFM) to unravel the genotoxic mechanisms of three different types of AuNPs, namely, small (~10 nm diameter) AuNPs (sAuNPs), large (~22 nm diameter) AuNPs (LAuNPs), and large conjugated (~39 nm diameter) AuNPs (LCAuNPs). The conjugation of AuNPs is commonly used in clinical studies to increase their biocompatibility and targeting [13]. Compared to other advanced microscopic methods like scanning electron microscopy and scanning tunneling microscopy, 4DAFM has distinct advantages. It allows imaging in the native environments of biomolecules with atomic resolution while preserving the integrity of the samples. Moreover, the sample preparation for 4DAFM is relatively straightforward compared to other techniques, making it a convenient tool for studying dynamic processes in near-molecular environments [14,15,16].

## 2. Results

Figure 1a shows an AFM image of individual DNA plasmids in 0.1× PBS containing 1 mM NiCl_2_ before the addition of AuNPs. Their average deconvoluted width and height were 4.2 ± 0.7 nm and 2.3 ± 0.5 nm, respectively. Figure 1b shows an AFM image of small non-conjugated AuNPs (sAuNPs) on mica, in 0.1× PBS. The image reveals individual AuNPs that are uniformly distributed. The average diameter of 122 AuNPs was 16.57 ± 3.74 nm, and the average contour height was 5.4 ± 0.7 nm. Figure 1c,d present the distributions of the diameters and mean contour heights of the sAuNPs, respectively.

Figure 2a–c show the effect of non-conjugated sAuNPs on individual open circular DNA in 0.1× PBS. The open circular structure of the plasmid was no longer seen, and instead, beads-on-a-string features were observed. The average width and height of the DNA strings without the bead-like structures were 11.6 ± 3.2 nm and 1.76 ± 0.53 nm, respectively. In contrast, the average width and height of the DNA strings that contain beads (red features) were 16.9 ± 5.5 and 2.5 ± 0.63 nm, respectively. The tracking length (240 nm) of Figure 2d is seen in Figure 2e. Figure 2f shows a cross-section of one bead (red color) depicted in Figure 2d. This cross-section provides a closer look at the structure and dimensions of the bead formed by the wrapping of DNA around the sAuNP. Additionally, Figure 2g shows the height distribution of the structures observed in Figure 2d, providing information about the variation in height within the DNA bead structures. Scale bar = 100 nm, and z scale = 4 nm.

Figure 3 presents a time-lapse series of AFM images illustrating the effects of sAuNPs on an individual DNA plasmid in 0.1× PBS over a period of 1 to 25 min. The scale bar = 50 nm. Initially, after 1 min of adding sAuNPs to the DNA molecules, the original open circular structure of the DNA plasmid was no longer visible. Instead, “beads-on-a-string” features were observed. After 7 min, the DNA molecule began to fragment at specific positions (labeled as a–d). The dynamic process of the effect of AuNP on DNA was quantitatively analyzed by measuring the length, width, and height of the DNA fragments, as well as the size of the DNA-AuNP beads. Specifically, the length of the b–c fragment decreased over time from 96.5 nm (at 0 min) to 86.9 nm (at 7 min) and finally to 73.4 nm (at 25 min). Simultaneously, the height of the same fragment increased from 2.57 nm to 2.74 nm and 2.91 nm at 0, 7, and 25 min, respectively. Furthermore, the width of the b–c fragment increased from 21 nm to 23 nm and 27 nm at 0, 7, and 25 min, respectively.

Figure 4 shows the effect of large AuNPs on individual DNA molecules in 0.1× PBS. Figure 4a shows LAuNPs. The average diameter and height of 209 nanoparticles were 26.81 ± 6.51 nm and 5.43 ± 0.17 nm, respectively. Aggregates of LAuNPs were also seen. Figure 4b,c depict their average diameter and height distribution, respectively. Figure 4d illustrates the effect of LAuNPs on single DNA plasmids. DNA fragments of all possible lengths are seen and continue to degrade toward the ultimate completeness of the DNA molecules. Figure 4d,e shows the average contour length and height distribution of the labeled molecule. Scale bar = 100 nm.

Figure 5 shows the effect of the large conjugated AuNPs (LCAuNPs) on individual DNA molecules in 0.1× PBS. Figure 5a displays an image of LCAuNPs. The average diameter of 98 LCAuNPs was 38.8 ± 10.64 nm, and the average contour height was 5.87 ± 1.38 nm. Similarly, the aggregates of LCAuNPs were seen. The LCAuNPs have smaller measured height dimensions than their measured diameters. Figure 5d shows the effect of LCAuNPs on a single DNA plasmid. Different possible oligonucleotide fragments are highlighted in Figure 5e–g. In Figure 5e, three voids of different lengths are seen within the DNA molecule. These lengths, from down left to upright, were 144.6 nm (~425.3 bp and ~42.5 turns), 36.2 nm (~106.4 bp and ~10.6 turns), and 108.5 nm (319 bp and ~32 turns). Figure 5f shows another DNA fragment (520 nm in length). The diameter of each of the two bright beads-like structures was 64.5 nm (190 bp and 19 turns), and the distance between these two beads was 176.5 nm (519 bp and 51.9 turns). Finally, Figure 5g shows a large gap of 674 nm (1983 bp and 198.3 turns). The contour lengths of Figure 5e–g are seen in Figure 5h–j.

## 3. Discussion

We presented an AFM study illustrating individual DNA plasmids, with a single molecule identified by its measured circumference (1223 ± 43 nm), compared to the theoretical contour length of the plasmid DNA (1484.1 nm). This representation was captured in the molecular environment prior to the introduction of AuNPs. The average deconvoluted width and height of the individual DNA molecules were 4.2 ± 0.7 nm and 2.3 ± 0.5 nm, respectively. Our findings align with reported DNA diameter variations (3.5 nm to 14 nm), with measurements less than 50% of the X-ray diffraction value [17,18].

Additionally, our AFM images displayed uniformly distributed sAuNPs, LAuNPs, and LCAuNPs, as well as aggregates, possibly influenced by factors such as electrostatic damping or salt bridging from the PBS solution. In the case of LCAuNPs, other factors such as the hydrogen bonding and interpenetration of conjugated peptide chains are probable contributors to aggregate formation. The observed height discrepancy of AuNPs, lower than their diameters, could be attributed to their insertion into salt layers on the mica surface due to the presence of PBS.

Introducing non-conjugated sAuNPs to an open circular DNA revealed beads-on-a-string features, most likely representing the initial DNA packaging around sAuNPs. This process effectively neutralizes the negative charges present in the DNA phosphodiester chain [19]. The electrostatic interaction at play contributes to alleviating DNA rigidity by minimizing repulsion between negatively charged phosphate groups through phosphate screening [20]. Additionally, the presence of condensed Na^+^ ions, derived from the PBS solution, within the negatively charged DNA base pairs, can contribute to the shielding of the coulomb repulsion between phosphate groups. This shielding effect is likely to cause the bending of the DNA molecules, leading to the formation of kinks or local base pair unstacking [6,20]. Furthermore, the local density of condensed Na^+^ ions is reported to be higher in the direction of bending compared to the opposite direction. This uneven distribution of ions supports the bending of DNA [21]. This scenario is analogous to the wrapping of DNA around histones in the nucleus, where DNA condenses and forms compact structures to fit within the limited space [19].

Moreover, our 4DAFM study of the effect of sAuNPs on individual DNA plasmids in 0.1× PBS showed a transition from an open circular structure to “beads-on-a-string” features in 1 min, followed by DNA fragmentation after 7 min, likely due to double-strand breaks (DSBs). The dynamic impact of AuNPs on DNA was quantitatively analyzed by measuring the length, width, and height of the DNA fragments, as well as the size of the DNA-AuNP beads. The observed increase in width and height of the fragments, along with the decrease in length, suggests that the DNA molecule experienced stress when interacting with the AuNPs. This stress-induced deformation is due to the wrapping of DNA around the AuNPs, resulting in geometric distortion or strain [22]. As the DNA wraps around the AuNPs, the entropy of the DNA molecule is reduced, leading to a contracted state and altered spring constant. When the stress is relieved through the formation of double-strand breaks (DSBs), the DNA filament returns to a state of higher disorder, and its original spring constant is restored, causing the filament to contract. The interaction between AuNPs and DNA significantly affects the trajectory of DNA molecules and alters Young’s modulus, which characterizes the stress–strain relationship.

The observed average circumference of sAuNPs (31.4 nm) suggested at least one turn of DNA double strands (~92 base pairs) around an individual sAuNP. Despite the length of this single turn being below the minimum recommendation of the conventional worm-like chain (WLC) model (150 bp), the WLC model remains applicable for DNA lengths as short as 85 bp (28.9 nm) [23]. It is noteworthy that the WLC model may not fully describe the elastic bending behavior of DNA filaments at shorter length scales (15–50 bp), as deviations from the model have been reported [24]. Yet, the bending resulting from DNA wrapping around AuNPs introduces deformations and disrupts DNA symmetry. The formation of kinks is postulated to localize these deformations, thereby lowering the energetic costs linked with bending. This process relieves elastic strain and enhances bidirectional torsional flexibility [6,25,26]. This behavior mirrors the scenario observed with nucleosomes. Therefore, the possibility of double-strand breaks (DSBs) occurring solely due to the stress formed by the physical wrapping of DNA around AuNPs can be ruled out. Instead, it is suggested that strong direct interactions may occur between the nitrogen bases of DNA and the gold surface, leading to the disruption of hydrogen bonds between complementary nucleotides and the separation of DNA into single strands [27]. Additionally, the gold surface itself may act as a Lewis acid, coordinating with phosphate oxygen in the DNA backbone and mediating the cleavage of phosphodiester bonds [2]. The combined effect of hydrogen bond breakage and phosphodiester bond cleavage can result in single-strand breaks (SSBs) and DSBs within DNA molecules.

We also explored the effects of LAuNPs and LCAuNPs on DNA, revealing fragments of all lengths that degrade, causing ultimate breakdown. The contact points between DNA fragments and AuNPs are influenced by the persistence length, impacting the subsequent decomposition process. Specifically, as the persistence length of a DNA fragment decreases, it increases the number of contact points with the surface of AuNPs [27]. Consequently, after the primary DNA molecules break down into smaller fragments, the decomposition process continues. The inner segments of DNA fragments are more rigid than those at the fragment ends of the same length because the base pairs at the ends of the fragments fluctuate with less spatial constraints and thus have greater bending/extension flexibility [28].

Ultimately, despite the robust nature of the thiol-Au bonds present on the LCAuNPs surface, it is noteworthy that prior investigations have documented the desorption and cleavage of these thiol groups from the AuNPs surface [29]. The release of these desorbed thiol groups may facilitate the liberation of free oxidized Au(I) atoms from the AuNPs surface [30]. It is probable that these liberated gold atoms, now in a free state, function as Lewis acids, coordinating with phosphate oxygen in the DNA backbone and participating in the mediation of phosphodiester bond cleavage [2].

Therefore, the genotoxic effects observed in this study are likely due to the direct interactions between DNA and the different types of AuNPs, including the breaking of hydrogen bonds and the cleavage of phosphodiester bonds, rather than only being a result of the physical stress caused by DNA wrapping around the AuNPs. Finally, the flexibility of DNA is dependent on its sequence, temperature, and ionic condition, which is beyond the scope of the present work.

## 4. Materials and Methods

All materials, unless stated otherwise, were purchased from Sigma-Aldrich (St. Louis, MO, USA) and used without further purification. Water and buffers were filtered before use through a 0.2 μm pore size filter (Sartorius, Gottingen, Germany).

### 4.1. Preparation of Gold Nanoparticles (AuNPs) 

#### 4.1.1. Non-Functionalized AuNPs

To confirm whether the size is a property that controls the toxicity of AuNPs in vitro, two types of non-functionalized AuNPs were used. Small non-functionalized AuNPs (sAuNPs), with average diameter of 10 nm, and large non-functionalized AuNPs (LAuNPs), with average diameter of 22 nm, were obtained from the Laser Dynamics Laboratory of Professor Mostafa A. El-Sayed at the Georgia Institute of Technology. These LAuNPs were synthesized by the citrate reduction method [31]. Both AuNPs were diluted with 1 mM phosphate-buffered saline (PBS), to prepare 0.1 nM solutions, and stored at 4 °C.

#### 4.1.2. Large Conjugated AuNPs (LCAuNPs)

The LCAuNPs were obtained from the Laser Dynamics Laboratory of Professor Mostafa A. El-Sayed at the Georgia Institute of Technology, with average diameter of 22 nm. The PEGylation and peptide conjugation were conducted according to the method reported by Austin, LA [32]. Briefly, The LAuNPs were PEGylated with thiol-terminated polyethylene glycol to achieve a 103-molar excess of PEG-SH. PEGylated LAuNPs were then purified and conjugated with arginine–glycine–aspartic acid (RGD) and nuclear localization signal (NLS) peptides at 104- and 105-molar excesses, respectively. LCAuNPs were then diluted with 0.1× PBS to make 0.1 nM solutions and stored at 4 °C.

### 4.2. Preparation of Naked DNA Solution

A stock solution (50 μg mL^−1^ *w*/*v*) of lyophilized DNA plasmid pBR322, 4365 bp, was made in deionized (DI) water and further diluted to 4 μg mL^−1^ before use in 0.1× PBS containing 1 mM NiCl_2_, pH 7.4.

### 4.3. AuNPs Preparation for AFM Imaging in PBS

A 30 μL volume of 0.1 nM AuNPs in 0.1× PBS was deposited onto freshly cleaved mica, left for 5 min, and directly imaged in the 0.1× PBS solution.

### 4.4. AFM Imaging of Individual DNA Molecules

In situ biomolecular imaging by AFM requires the molecules to be sufficiently mobile to allow their dynamic behaviors to be monitored, and at the same time, the molecule must remain attached to the surface to allow good imaging [33]. In view of this, our method was developed to allow individual DNA molecules to bind to mica substrate via the application of NiCl_2_ (divalent cation) in 0.1× PBS to provide gentle bridges between DNA molecules and the mica substrate (both the DNA and mica are negatively charged at the pH 7.4). A 30 µL volume of the naked DNA solution was then introduced into the mica substrate, left for 5 min, and imaged to determine its integrity in the absence of AuNPs.

### 4.5. DAFM Imaging of the Effect of AuNPs on Individual DNA Molecules 

A 30 μL volume of the bare DNA solution was added to 30 μL of 0.1 nM solution of AuNPs, manually mixed for a few seconds, and directly imaged in 0.1× PBS to study their effect on the morphology and integrity of the DNA molecules in real time.

### 4.6. Atomic Force Microscopy 

All AFM images were taken with Bioscope Catalyst-AFM (Bruker, Santa Barbra, CA, USA) mounted on Lica inverted microscope (Wetzlar, Germany). Silicon nitride sharpened probes were used with nominal tip radius R = 2–12 nm (Bruker, Santa Barbra, CA, USA). Each probe has a force constant of 0.06–0.04 N.m^−1^, measured by the thermal tuning method. Deflection sensitivity was determined on glass. Tips were cleaned in Piranha solution (70/30 H_2_SO_4_/H_2_O_2_) for about 5 s, washed with DI water, and dried with a gentle stream of nitrogen just before imaging to free them from contamination. All images were conducted in Peak Force Quantitative Nanomechanical (PFQNM) mode. Parameter limits were optimized to exclude situations that produce significant artifacts, such as sample/tip damage or very noisy data sets. Scan rates of 1–2 Hz were applied. Freshly cleaved mica (Agar Scientific, Essex, UK) was first scanned in DI water to confirm the absence of contamination.

### 4.7. Data Analysis 

Unless otherwise indicated, post-imaging analysis was performed on Nanoscope 8.1 software (Bruker, Santa Barbra, CA, USA). Images were flattened to remove sample tilt. Molecular widths were taken at half the height of the peak, and heights were taken from the average baseline to the highest point of the peak [34].

## 5. Conclusions

Our study represents a significant advancement in understanding the intricate interactions between gold nanoparticles and DNA, particularly focusing on the direct time-dependent DNA damage without the presence of ROS. We utilized AFM imaging and real-time molecular dynamics to observe and analyze the events occurring in the molecular environment. The results not only support previous studies indicating time-dependent DNA damage caused by AuNPs [12,35] but also surpass those studies by providing the direct observations of alterations in DNA morphology and spontaneous dynamics within the molecular environment.

This approach allowed for the calculation of molecular dimensions, including those of DNA, DNA fragments, and DNA-AuNP complexes. Furthermore, the study suggests potential biomedical applications, highlighting gold nanoparticles’ role as antiviral and anticancer nano-targets due to their ability to damage nucleic acids and exhibit photothermal properties. The insights gained from the direct observations of AuNPs’ effects on DNA could further advance these applications.

## Figures and Tables

**Figure 1 ijms-25-00542-f001:**
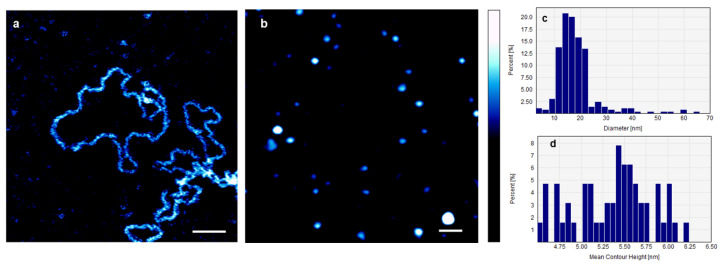
(**a**) AFM image of individual DNA plasmids in 0.1× PBS containing 1 mM NiCl_2_ before the addition of AuNPs. Their average deconvoluted width and height were 4.2 ± 0.7 nm and 2.3 ± 0.5 nm, respectively. (**b**) AFM image of AuNPs on mica, in 0.1× PBS. The image shows individual uniformly distributed AuNPs. The average diameter of the AuNPs = 16.57 ± 3.74 nm, and their average contour height = 5.4 ± 0.7 nm. Aggregates of AuNPs are also visible in the image. (**c**,**d**) Illustration of the distributions of the average diameters and the average contour heights of AuNPs, respectively. Scale bar = 100 nm, and the color bar = 10 nm.

**Figure 2 ijms-25-00542-f002:**
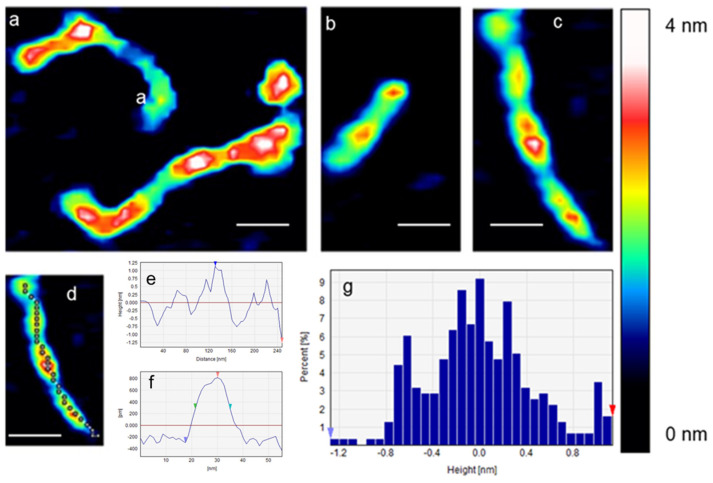
(**a**–**c**) AFM images of the effect of sAuNPs on individual DNA plasmids in 0.1× PBS. The tracking length of (**d**) is seen in (**e**), and the width and height of the cross-section of (**d**) are seen in (**f**). The height distribution of (**d**) is seen in (**g**). Scale bar = 100 nm.

**Figure 3 ijms-25-00542-f003:**
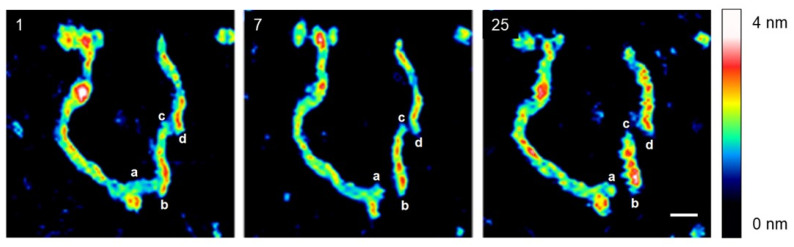
The time-lapse sequence (1–25 min) of AFM images capturing the effect of AuNPs on an individual DNA plasmid in a 0.1× PBS solution. Initially, after 1 min of adding AuNPs to the DNA solution, beads-on-a-string features were observed, likely due to DNA packaging around the AuNPs. Subsequently, at 7 min, the DNA molecule began to fragment at specific positions (a–b and c–d), possibly indicating the formation of double-strand breaks (DSBs). Over time, the length of the b–c fragment decreased, while its height and width increased, suggesting that the DNA molecule was experiencing stress.

**Figure 4 ijms-25-00542-f004:**
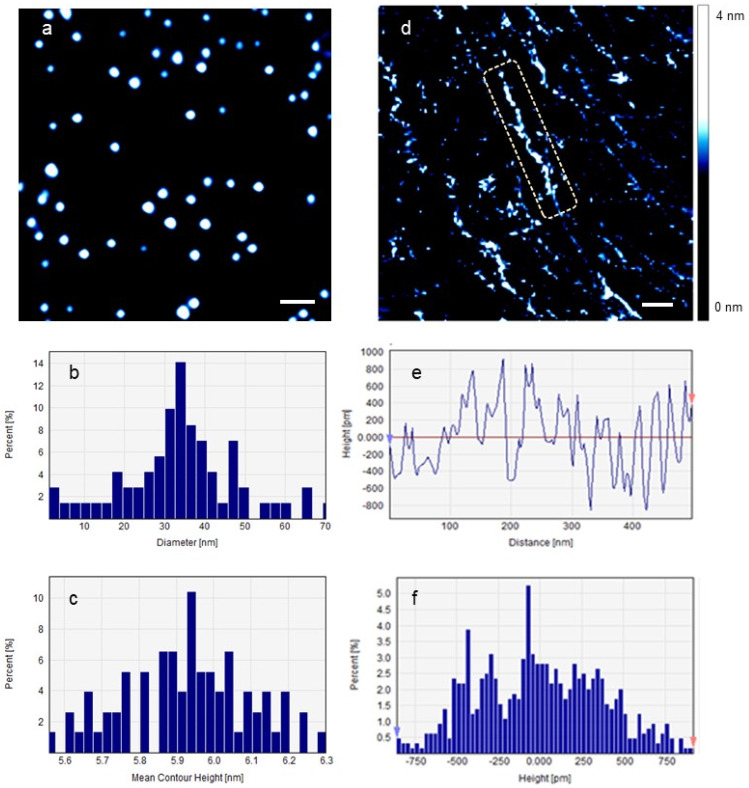
(**a**) LAuNPs, (**b**,**c**) the diameter and contour height of LAuNPs, respectively, (**d**) the effect of LAuNPs on DNA plasmids imaged on freshly cleaved mica in 0.1× PBS, the dashed rectangle highlights a substantial segment of the damaged DNA molecule, and (**e**,**f**) the average contour length and height of the labeled molecule, respectively. Scale bar = 100 nm.

**Figure 5 ijms-25-00542-f005:**
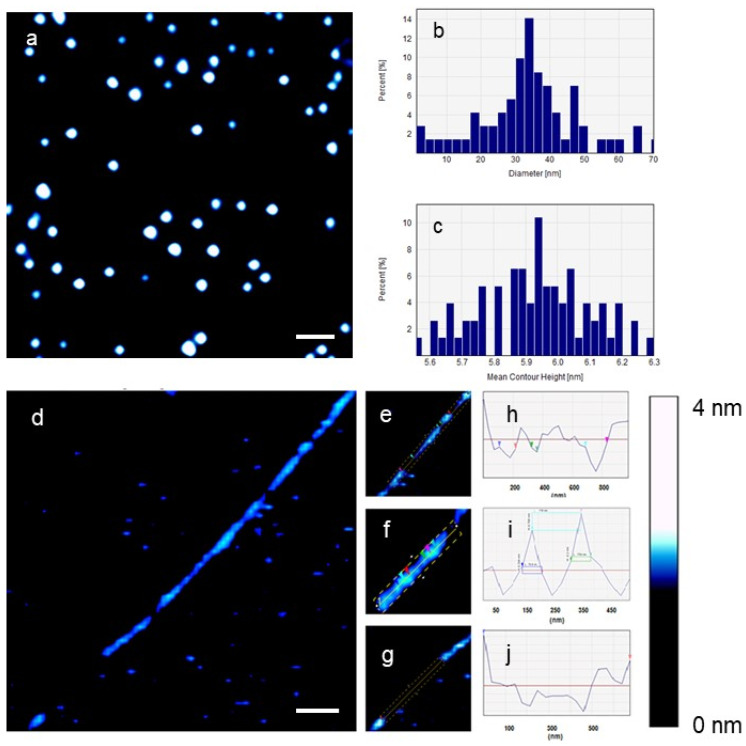
The effect of LCAuNPs on a single DNA plasmid imaged on freshly cleaved mica in 0.1× PBS. (**a**) LCAuNPs, (**b**,**c**) the diameter and contour height of LCAuNPs, respectively, (**d**) the effect of LCAuNPs on a single DNA plasmid, (**e**–**g**) are different fragments of the DNA molecule depicted in (**d**), the dashed rectangle in (**f**) shows the interaction between a DNA fragment and two AuNPs, and (**h**–**j**) the average contour length and height of the labeled molecular fragments, respectively. The colored arrowheads in subfigures (**h**–**j**) correspond to those in subfigures (**e**–**g**), respectively. Scale bar = 300 nm.

## Data Availability

The raw data necessary for replicating these results can be obtained by reaching out to the corresponding author. Comprehensive details of all methods employed in this study are provided in the Section 4.

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
