# Peer review of "Visualizing the 4D Impact of Gold Nanoparticles on DNA"

_ijms, 2023, doi:10.3390/ijms25010542_

Round 1
Reviewer 1 Report
Comments and Suggestions for Authors
AuNPs has been widely concerned in sensing, drug delivery and therapy in biomedical fields. At the same time, the safety of AuNPs has been also concerned. Therefore, the study of mechanism on the interaction between nanogold and biomolecules has become a particularly important research topic. This manuscript reported some results regarding the visualizing the 4D Impact of AuNPs on DNA, indicating that there were the significantly different DNA chain morphology caused by different particle size and surface modification. These results are important to some extent for the study of AuNPs interaction with DNA. However, in my opinion, authors should consider the following problems.
(1) This study can only reflect the results of the morphology changes, and it is difficult to speculate on the reasons for these changes. Therfore, the conclusion should be narrowed narroded down to the significant change in morphology. That is, AuNPs with different sizes and surface modifications does have a significant impact on DNA morphology.
(2) There is a certain electric field at the tip of the atomic force microscope, and whether the interaction between the charged AuNPs (the surface of AuNPs always modified with charged ions) and DNA could be affected by this electric field. In other words, whether the results obtained in this investigation are dependent on AFM. In fact, such condition has been confirmed in the literature. (Effects of alternating electric field on the imaging of DNA double-helix structure by atomic force microscope. Appl Nanosci 10, 3517–3524 (2020)). Thus, the electric field action at nano scales may be a worthwhile consideration, especially in case of presence of AuNPs with different charged features and sizes. When attempting to observe biological processes in vitro with AFM, it is essential to ensure that the measurement only minimally perturbs the interactions of interest.
(3) More valuable results may be obtained if authors could observe the results in an aerobic or anaerobic atmosphere environment and combine them with reactive oxygen species detection methods.
Author Response
Reviewer:
- This study can only reflect the results of the morphology changes, and it is difficult to speculate on the reasons for these changes. Therefore, the conclusion should be narrowed narroded down to the significant change in morphology. That is, AuNPs with different sizes and surface modifications does have a significant impact on DNA morphology.
Reply:
Thank you for your thorough review and valuable feedback, which has significantly improved the focus and clarity of our study. In response to your suggestion to narrow down the conclusion to the significant changes in morphology, we have made the following addition to the conclusion section
The results not only support previous studies indicating time-dependent DNA damage caused by AuNPs [13,35] but also surpass those studies by providing direct observations of alterations in DNA morphology and spontaneous dynamics within the molecular environment. lines 297-300.
Reviewer:
(2) There is a certain electric field at the tip of the atomic force microscope, and whether the interaction between the charged AuNPs (the surface of AuNPs always modified with charged ions) and DNA could be affected by this electric field. In other words, whether the results obtained in this investigation are dependent on AFM. In fact, such condition has been confirmed in the literature. (Effects of alternating electric field on the imaging of DNA double-helix structure by atomic force microscope. Appl Nanosci 10, 3517–3524 (2020)). Thus, the electric field action at nano scales may be a worthwhile consideration, especially in case of presence of AuNPs with different charged features and sizes. When attempting to observe biological processes in vitro with AFM, it is essential to ensure that the measurement only minimally perturbs the interactions of interest.
Reply:
We would like to clarify that our AFM imaging was conducted without the influence of an external electric field. Unlike the study mentioned in the literature that imaged DNA under different alternating electric fields and intensities to induce DNA stretching and unfolding, our AFM imaging specifically focused on probing the direct interaction between the AFM probe and the sample without the presence of any external electric field. The absence of an external electric field in our experimental setup ensures that we are solely examining the direct interactions of interest without additional perturbations from electric field effects.
Reviewer:
(3) More valuable results may be obtained if authors could observe the results in an aerobic or anaerobic atmosphere environment and combine them with reactive oxygen species detection methods.
Reply:
To address the reviewer's suggestion, we acknowledge the importance of exploring results in both aerobic and anaerobic environments and combining them with reactive oxygen species (ROS) detection methods. Our ongoing research has already demonstrated that AuNPs can induce DNA damage even in the absence of ROS within a molecular environment. This valuable insight will be taken into consideration and integrated into our upcoming experiments, which will include observations in both aerobic and anaerobic atmospheres, along with comprehensive ROS detection methods.
Reviewer 2 Report
Comments and Suggestions for Authors
In this work visualizing the 4D impact of AuNPs on DNA is described. Authors employed four-dimensional atomic force microscopy to examine the ability of AuNPs to damage DNA in vitro in the absence of ROS. To further examine whether the size and chemical coupling of these AuNPs are properties that control their toxicity, authors exposed individual DNA molecules to three different types of AuNPs: small, large, and large conjugated AuNPs. The findings of this work holds significant promise for advancing nanomedicines in diverse areas like viral therapy, cancer treatment, and biosensor development for detecting DNA damage or mutations. The article looks like a Communication and may be published after minor revision.
Notes:
1. The axis labels of Fig. 1 c), d) and Fig. 2 e), f) should be increased for clarity.
2. There are not enough additional methods to confirm DNA damage after AuNPs adding. The charges change present in the DNA chain after adding of AuNPs should be confirmed by the measurement of zeta potentials by dynamic light scattering method. The deformations and disrupts DNA symmetry should be confirmed by measuring CD spectra (circular dichroism).
Author Response
Reviewer
1- The axis labels of Fig. 1 c), d) and Fig. 2 e), f) should be increased for clarity.
Reply:
We appreciate your in-depth review and valuable feedback, contributing significantly to the enhanced focus and clarity of our study.
We have increased the axis labels of Fig. 1 c), d) and Fig. 2 e), f) for clarity. And we have incorporated them into the manuscript.
Reviewer:
2- There are not enough additional methods to confirm DNA damage after AuNPs adding. The charges change present in the DNA chain after adding of AuNPs should be confirmed by the measurement of zeta potentials by dynamic light scattering method.
Reply:
Typically, concentrations within the range of 1-10 mg/mL are considered suitable initial values for determining the optimal sample concentration in DLS measurements. However, in our case, we are working with minimal amounts, specifically a few micrograms of DNA (4 µg) and 0.1nM of AuNPs, which fall below the zeta potential detection limit. We intend to employ low concentrations to observe the impact of a small number of AuNPs on individual DNA strands. This is why we have chosen our AFM imaging method, along with 4DAFM imaging, as quick and straightforward techniques to directly illustrate (seeing is believing) the influence of AuNPs on DNA, eliminating the necessity for additional methods.
Reviewer:
3- The deformations and disrupts DNA symmetry should be confirmed by measuring CD spectra (circular dichroism).
Reply:
We acknowledge the significance of verifying deformations and disruptions in DNA symmetry through CD spectra (circular dichroism). In our upcoming research, we intend to utilize CD spectroscopy for a thorough investigation and clarification of these structural changes. We aim to provide a comprehensive analysis that ensures a detailed examination of interactions and deformations with high sensitivity and precision. In the present communication, our emphasis lies in demonstrating conclusively, through AFM, that AuNPs can cause DNA damage in the absence of ROS.
Reviewer 3 Report
Comments and Suggestions for Authors
In the manuscript Abdelhady et al. describes the 4D Impact of AuNPs on DNA. Manuscript is compiled well with nice experimental discussions. There are few points authors should discuss before considering it for publication in IJMS.
1. Authors have discussed about size of GNPs but how about the concentration or biodistribution of GNPs affect DNA damage?
2. Authors should arrange a table for size and other factors with time dependent DNA damage.
3. What is the role of laser beam in the DNA damage during AFM study?
4. Does the single or double strand break depend on the size of GNPs. Authors should explain if they have observed any scenario.
5. Authors should provide the reference/s for the statement “The conjugation of AuNPs is commonly used in clinical studies to increase 53 their biocompatibility and targeting.” In introduction line number 53.
Author Response
Dear reviewer, your extensive review and valuable insights are deeply appreciated, greatly contributing to the refinement and clarity of our study. Our replies are attached. Thanks much.
